# Impact of four years of annually repeated indoor residual spraying (IRS) with Actellic 300CS on routinely reported malaria cases in an agricultural setting in Malawi

**Remy Hoek Spaans**[1]*, **Albert Mkumbwa**[2], **Peter Nasoni**[2], **Christopher M. Jones**[2,3☯], **Michelle C. Stanton**[2☯]

**1** Department of Vector Biology, Liverpool School of Tropical Medicine, Liverpool, United Kingdom, **2** Illovo Sugar Malawi, Nchalo, Malawi, **3** Malawi-Liverpool-Wellcome Trust, Blantyre, Malawi

☯ These authors contributed equally to this work.
* remyhoekspaans@gmail.com

## Abstract

Indoor residual spraying (IRS) is one of the main vector control tools used in malaria prevention. This study evaluates IRS in the context of a privately run campaign conducted across a low-lying, irrigated, sugarcane estate from Illovo Sugar, in the Chikwawa district of Malawi. The effect of Actellic 300CS annual spraying over four years (2015-2018) was assessed using a negative binomial mixed effects model, in an area where pyrethroid resistance has previously been identified. With an unadjusted incidence rate ratio (IRR) of 0.38 (95% CI: 0.32–0.45) and an adjusted IRR of 0.50 (95% CI: 0.42-0.59), IRS has significantly contributed to a reduction in case incidence rates at Illovo, as compared to control clinics and time points outside of the six month protective period. This study shows how the consistency of a privately run IRS campaign can improve the health of employees. More research is needed on the duration of protection and optimal timing of IRS programmes.

## Introduction

Malaria is a life-threatening disease that affects 247 million people globally, with 95% of the case burden falling on the African region as defined by the World Health Organization (WHO) [1]. The two most impactful vector control tools at a national malaria control programmes' (NMCP) disposal are insecticide-treated nets (ITNs) and indoor residual spraying (IRS) [2, 3]. During IRS campaigns the indoor walls and sometimes roofs of houses are treated with insecticides. Only a small number of insecticide classes are used to treat ITNs, which combined with prolonged use of IRS using these same classes, has led to the development of insecticide resistance [4]. Compared to ITNs, there is a larger range of insecticides classes used for IRS, with different modes of action (organochlorines, organophosphates, carbamates, pyrethroids, neonicotinoids, and pyrroles) [5]. Whilst ITNs are ubiquitously used by NMCPs, IRS using predominantly the insecticide dichlorodiphenyltrichloroethane (DDT) was favoured over the period 1955–1969 by the Global Malaria Eradication Programme (GMEP), after

**Data Availability Statement:** All datasets and metadate required to reproduce the analysis are

published open access on Zenodo. Access via: 10.5281/zenodo.10626879.

**Funding:** This work was funded by the Medical Research Council through the Medical Research Council Doctoral Training Partnership (1965090) awarded to Remy Hoek Spaans through the Liverpool School of Tropical Medicine and Lancaster University. The funders had no role in study design, data collection and analysis, decision to publish, or preparation of the manuscript.

**Competing interests:** I have read the journal's policy and the authors of this manuscript have the following competing interests: Dr. Albert Kumbwa is employed by Illovo Sugar as the main clinician at Illovo Sugar Nchalo. Peter Nasoni was employed as the public health officer at the time of data collection.

which it fell out of favour as a malaria control tool [6, 7]. Although its use has been increasing again over the past 20 years, the delivery of IRS programs remains patchy and inconsistent [8]. IRS is a logistically demanding and relatively expensive intervention, that requires long-term commitment in terms of funding, procurement, and training of the spray team [9]. Therefore, it is mostly used as a spatially-targeted intervention in high burden, densely populated areas [10]. For practical reasons and in anticipation of the malaria season, IRS campaigns are usually planned to finish close to the start of the wet season [11]. Whilst hut trials leave little doubt about the *efficacy* of IRS under controlled conditions [12–14], this data has limited generalisability [15]. The *effectiveness* of IRS on both entomological and malaria outcomes depends on variations in the local environment, which considering the long history of IRS in malaria control, are poorly understood [3, 16].

IRS targets mosquitoes resting on the walls and surfaces inside houses, which will either be killed or will have a reduced life span after exposure to the insecticide. Therefore, any effects observed in reduced malaria case incidence rely on the assumption that malaria transmission occurs mainly indoors. This is increasingly less so in Sub-Saharan Africa (SSA) with evidence suggesting that *Anopheles spp.* may be shifting towards more exophagic behaviour [17–19]. Furthermore, effectiveness of IRS depends on household coverage of the study area, spray quality, and residual activity of the insecticide used. For example, the residual activity of the organophosphate Actellic 300CS in optimal conditions can be up to nine months [13]. However, when President's Malaria Initiative (PMI) field sites were compared across 17 countries, a wide range of two to nine months of residual activity was found for pirimiphos-methyl [15]. This discrepancy can be explained by a range of factors such as the final dosage of insecticide on the wall, interaction with wall surface type, adaptations to the house after spraying, and environmental conditions such as temperature and humidity [15, 20, 21]. With increased ITNs coverage over the past decade, IRS is now more frequently used alongside ITNs, rather than as an alternative [22]. There is conflicting evidence on the added benefit of IRS when combined with ITNs. Clinical cluster-randomized trial data suggests that there is some impact of non-pyrethroid-like IRS co-deployed with ITNs, but the evidence is inconsistent [16]. WHO recommends that IRS and ITN should only be combined when different insecticide classes are used [3].

In Malawi, the number of households with at least one ITN has steadily increased from 55% to 82% from 2012 to 2017 [23]. Mass distribution campaigns have occurred in 2012, 2014, 2016, 2018, and 2021. The Malawi Malaria Strategic Plan (MSP) 2011–2016 put forward a phased roll-out of Rapid Diagnostic Testing (RDT's) and training of Health Surveillance Assistants (HSAs) [24]. Indoor Residual Spraying (IRS) implementation and coverage in the country has been patchy with a high reliance on external funding [25]. Despite the detection of widespread insecticide resistance to carbamate and pyrethroid resistance in 2010, due to financial constraints, in 2012 pyrethroids were still used in six districts and organophosphates in one district in Malawi [8, 25]. After this period, nationally organised IRS was scaled down and stopped completely in 2016–2017. The MSP 2017–2022 reintroduced plans to use targeted IRS in areas with high transmission intensity with the aim to scale up to 11 districts by 2022 [26]. Since 2017, IRS coverage was scaled up to four districts in 2020–2021 with the support of PMI and the Global Fund. Three different insecticides were used: Actellic 300CS (organophosphate), SumiShield 50WG (neonicotinoid), and Fludora Fusion (mixture of pyrethroid and neonicotinoid) [25]. A recent President's Malaria Initiative (PMI) report revealed that the residual life of Actellic 300CS in Malawi varied between two and five months [25].

Several less-documented, smaller-scale IRS campaigns are conducted by the private sector in Malawi [24, 27]. One of the longest privately run routine IRS campaigns is run by the Malawian branch of the Illovo Sugar Africa company with two locations in central (Nkhotakota)

and southern (Chikwawa) districts [27]. The Illovo site in Chikwawa (Nchalo estate) sits within the low-lying Shire Valley in the south of Malawi, where the main malaria vectors are *An. arabiensis* and *An. funestus* [18]. The use of irrigation systems with year-round provision of water bodies, provides an ecological niche for local vectors throughout the dry season [28–31]. IRS was implemented in 1990 but by 2014, low level resistance against the pyrethroid deltamethrin was detected, with WHO cone assays detecting 87% mortality (n = 791) [32]. After these findings the IRS programme switched the active ingredient of the insecticide from a pyrethroid (alpha-cypermerthrin) to an organophosphate (Actellic 300CS). Vectors in the area remain fully susceptible to pirimiphos-methyl [33]. The on-site malaria health records collated by Illovo combined with coverage data from the routine IRS provide an opportunity to investigate the impact of IRS on local clinical malaria cases. This study investigates the impact of IRS using Actellic 300CS on malaria cases reported at seven clinics over a four year period (2015–2018), compared to three control clinics outside of the estate and time periods outside of the protective period.

## Materials and methods

### Illovo study site and population

The Illovo Nchalo estate is located within the Shire Valley, Chikwawa district, in the Southern Region of Malawi (-16.258539, 34.890956) (Fig 1). The Shire Valley has a unique climatic zone within Malawi characterized by hot and humid conditions, with mean monthly temperatures

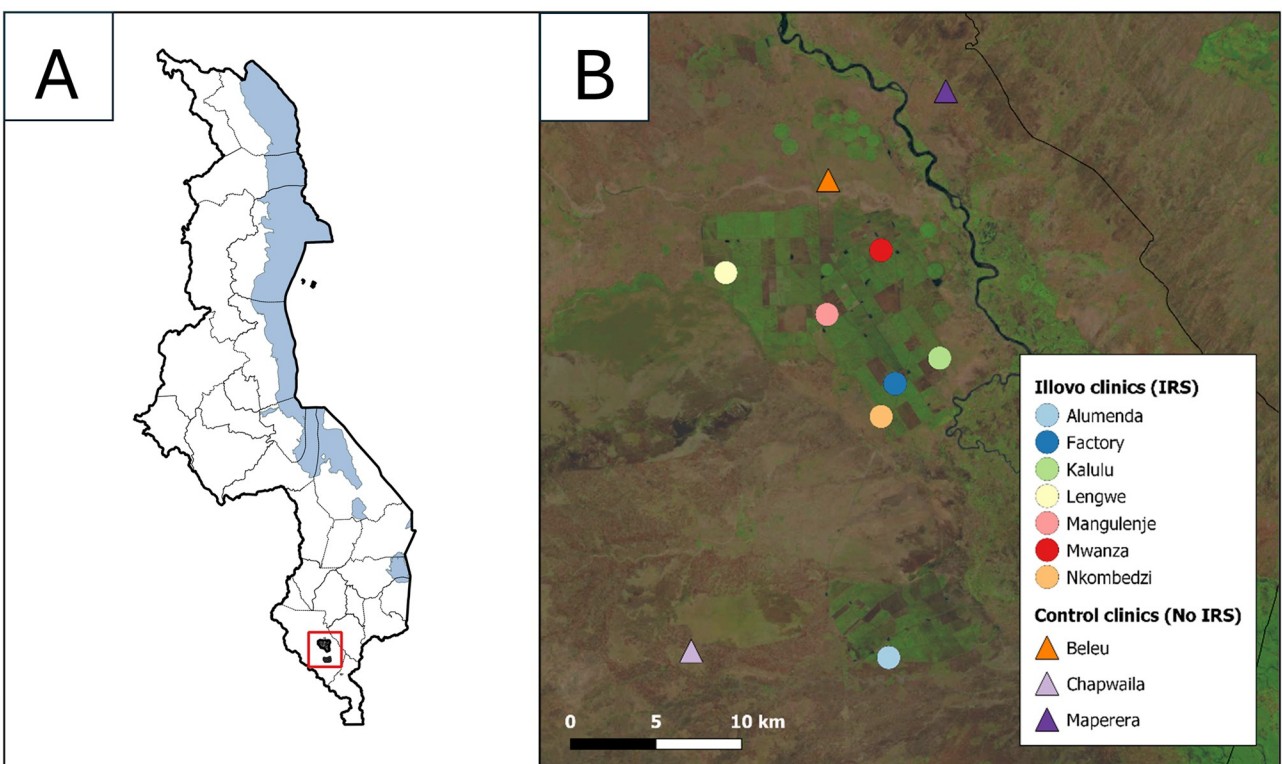

**Fig 1. Study area.** A) Location of the Illovo Nchalo Estate within Malawi. Administrative boundaries courtesy of GADM database of Global Administrative Areas and Regional Centre for Mapping of Resources for Development (RCMRD) [37, 38]. B) Agricultural fields of Illovo Sugar Nchalo. The seven Illovo health clinics that received IRS are indicated by circles. The three control clinics that did not receive IRS are indicated by triangles. The basemap was generated using a Landsat-8 image from 2015–11-14, courtesy of the U.S. Geological Survey [39].

between 19–26˚C and a single rainy season within the period November-April [34–36]. The estate covers over 150 km2 of sugar cane fields, a perennial crop that is grown as a monoculture, growing between 2–6m high. The Shire river that flows along the eastern boundary provides water supply for the irrigation system, a combination of centre pivot, sprinklers, drip -and flood irrigation. The main vectors are *An. arabiensis* and *An. funestus*, although *An. gambiae* sensu stricto and *An. quadriannulatus* are also present [19, 36]. A sugarcane processing factory, thirteen villages and seven out-patient (OPD) health clinics are present within the estate. The clinics provide free care to Illovo employees and their family members, registered as dependants. In fact, dependants make up the largest proportion of the Illovo population (80.0%), followed by employees (19.5%) with a permanent contract, and seasonal workers on a temporary contract (0.53%). As determined by the 2019 census, the on-site living population was 13,534, of which 47.7% was female. The IRS programme started in 1990 using bi-annual pyrethroid spraying, until resistance was detected in 2014 [32]. Starting in 2015, a switch was made to yearly spraying with the long-lasting formulation of organophosphate (pirimiphos-methyl), Actellic 300CS. A spray team consisting of 10 sprayers is trained each year and villages are incrementally sprayed between May and November.

## Data collection Illovo clinics

At Illovo, monthly malaria case data are routinely collected from out-patient registers at each clinic, aggregated, and entered into Malawi's District Health Information System (DHIS2) [40]. For Illovo's own records, a differentiation is made between on-site and off-site living patients, before entry into DHIS2. For this analysis only on-site living cases were included, because off-site villages were not covered by the IRS campaign. Illovo does not have in-patient facilities, therefore cases reflect RDT and microscopy positive out-patient cases. Every three years the on-site living population is enumerated by a census in February. An Illovo agricultural surveyor maps the area every year, for maps generated here, 2015 data was used. Data are reported for each village, by sex, age group, and employment status (Illovo/dependant/contractor). Data from the 2016 and 2019 census were used to estimate the expected population for 2015–2018, assuming linear growth. Daily weather data were obtained from the on-site manual weather station for the period 1999–2019; humidity (measured at 14:00), temperature (min, max, dry bulb measured at 14:00), and rainfall (mm per day).

## Selection of control clinics

To control for other factors that could explain reductions in malaria incidence over time (e.g. government mass bed net distributions), three control clinics with catchment areas that are not covered by IRS, were chosen. Clinics were selected based on type of health facility (out-patient clinic), distance to the boundary of the Illovo fields (<10km), completeness of data (>90%), and elevation (<100m difference to Illovo fields' maximum elevation). The clinic locations and the surrounding environments are indicated in Fig 1. The shortest distance to Illovo was 0.92km for Beleu, 5.63km for Maperera, and 7.89km for Chapwaila. Malaria data for control clinics was available through DHIS2. The indicator used was "WHO NMCP P Confirmed malaria cases", which combines outpatient RDT confirmed cases, outpatient microscopy confirmed cases, in-patient confirmed cases, and RDT positive case data from smaller village clinics within the clinic's catchment area.

## Data analysis

Monthly malaria case incidence was modelled using a negative binomial mixed effects model with an offset for the expected population size, including a random intercept for clinic

catchment area and a random slope for time. The aim of the analysis was to estimate the incidence rate ratio (IRR) for the effect of IRS by comparing locations and periods where IRS was implemented and effectively killing mosquitoes to locations and time points where this was not the case. The duration of the protective period afforded by Actellic 300CS was assumed to be 6 months [14, 15].

$$Y_{it}|\lambda_{it}, \kappa \; NegBin(\kappa, \lambda_{it}) \tag{1}$$

Where $Y_{it}$ is the observed count of malaria cases at clinic $i$ and time step in months $t$, given the rate $\lambda_{it}$ of cases and $\kappa$ the over-dispersion parameter.

$$\log\{\lambda_{it}\} = \alpha + f(t) + \phi IRS_{it} + (1 + t)U_i \tag{2}$$

$$U_i \sim N(0, v^2) \tag{3}$$

Where $f(t)$ represents a temporal trend on a monthly scale, $IRS_{it}$ represents IRS coverage for clinic $i$ in month $t$, $t = 1, \ldots, 48$, and $U_i$ represents clinic-level random effects. A random intercept for clinic with a random slope for time have been incorporated to allow changes in between-clinic variability over time to be captured.

Time series of the individual clinics showed an annually recurring seasonal pattern in malaria transmission. The temporal trend was decomposed into three components such that;

$$f(t) = \beta t + s(t) + w(t) \tag{4}$$

The $t$ term accounts for linear changes in incidence over time due to unmeasured covariates. Periodic fluctuations in malaria incidence i.e. seasonality were captured by the $s(t)$ term, and $w(t)$ represents deviations from the seasonal pattern as driven by local weather condition.

Generalized Linear Mixed Models (GLMMs) with a single fixed effect including IRS and weather-related covariates were initially fitted to reduce the number of possible model formulations and were assessed by Akaike Information Criteria (AIC), Root Mean Square Error (RMSE), correlation between predicted and observed cases, and mean residuals. A GLMM representing the seasonal component was fitted and later combined with the weather models. Seasonality was incorporated in the model by including a cosine function into the model where the amplitude $A$ and horizontal shift $\theta$ are estimated, for a specified period (T) over a number of time steps t. This is referred to as harmonic regression [41, 42]. In this case the outcome is measured in time steps of 1 month and period (T) is a year (12t).

$$s(t) = A * cos\left[\frac{2\pi t}{T} - \theta\right] \tag{5}$$

In order to estimate $A$ and $\theta$ through regression, the equation needs to be transformed to:

$$s(t) = \gamma_1 * sin\left(\frac{2\pi t}{T}\right) + \gamma_2 * cos\left(\frac{2\pi t}{T}\right) \tag{6}$$

A plot of the fitted harmonic regression line and methods to obtain ($\alpha$) and ($\theta$) can be found in Fig B in S1 File. Weather variables are incorporated into the model as anomalies, i.e. deviations from the long-term trend over a 20-year time period of weather station data. Daily measured variables were first aggregated by taking the mean over each month. After calculating anomalies, each weather variable was lagged by 1 to 3 months to reflect the time it takes for climatic variables to affect malaria cases through mosquito survival, mosquito abundance, malaria transmission intensity and time to diagnosis. The best fitting uni-variate weather models were combined with a model that included the seasonal component for further model

selection. The best models resulting from this process were combined with the IRS model. The previous month's malaria cases were added to the model as an auto-regressive term to check if it would improve model fit in case there was any additional non-seasonal temporal variation.

Cross-validation was done by randomly splitting the data into a hold-out and training set in a 20–80% split, which was repeated 10 times to calculate the CV-RMSE. Data were analysed using R version 3.6.3. Packages used for data processing, formatting, and plotting were part of the "tidyverse", mainly "ggplot2" and "dplyr". [43]. The "lubridate" package was used for date formatting [44]. The "lme4" package was used for model fitting using the "glmer.nb" function [45]. Prediction intervals were calculated using the "bootMer" function from "lme4", using 1000 simulations, bootstrapping conditional on the random effects. The "imputeTS" package was used for linear interpolation of weather data, where it was missing (1 month for relative humidity) [46]. Data and code available via DOI:10.5281/zenodo.10626879.

The model accounting for malaria seasonality and time was a better fit than the model accounting for IRS as assessed by AIC, RMSE, and CV-RMSE (Table 1). This emphasises the importance of adjusting for seasonality. The best combination of the seasonal model with weather variables included rain anomalies lagged 3 months (mm), maximum temperature anomalies lagged 1 month, relative humidity anomalies, and minimum temperature anomalies lagged 1 month. This model, representing the modelled temporal variation for each of the clinics, was then combined with the IRS variable (Table 1, model 5). Weather variables were removed one by one to see how this would affect model fit and minimum temperature anomalies lagged by 1 month was dropped from the model. Although the effect of maximum temperature anomalies from the preceding month on malaria incidence rate was borderline significant, it was still included in the model because it improved model fit. The last constructed model, which includes a variable for malaria cases of the previous month to account for temporal auto-correlation, had a slightly improved AIC, but the RMSE and generalisability as assessed by CV-RMSE calculated from the model validation decreased. Therefore the final model selected was model 5 from Table 1. The model formulation is described in Eq 7.

$$\log(\lambda_{ct}) = \alpha + \beta t + s(t) + w(t) + \psi IRS_{it} + (1 + t)U_i \tag{7}$$

**Table 1. Model comparisons.**

| | Covariates | AIC | RMSE | CV-RMSE |
|---|---|---|---|---|
| 1 | No covariates | 5115 | 173.31 | 182.93 |
| 2 | IRS | 5033 | 172.76 | 182.48 |
| 3 | Seasonal | 4960 | 151.48 | 157.32 |
| 4 | Seasonal + weather | 4941 | 150.17 | 162.46 |
| 5 | Seasonal + weather + IRS | 4883 | 139.69 | 151.77 |
| 6 | Seasonal + weather + IRS + $cases_{t-1}$ | 4870 | 148.07 | 162.29 |

All models are negative binomial models with a random intercept for clinic, a random slope for time, and an offset for population size. Models 3–6 also include time (months) as part of the seasonal component of the model as well as a harmonic regression component as described earlier in Eq 6 (Fig B in S1 File). The selected weather variables for models 5 and 6 included: rain anomalies lagged 3 months (mm), maximum temperature anomalies lagged 1 month, and relative humidity anomalies. Model 4 included a fourth weather variable (minumum temperature anomalies lagged 1 month) which was dropped after inclusion of IRS in model selection for model 5, because removal slightly improved model fit. AIC: Akaike Information Criterion, RMSE: root mean square error, CV-RMSE: mean RMSE for the hold-out set after 10 repeats or random 20–80% splits.

Where $\alpha$ is the intercept, $\beta * t$ represents the coefficient for the long-term time trend, $s(t)$ is expressed in Eq 6, and $w(t)$ is expressed in Eq 8. $vIRS_{it}$ represents the regression coefficient of the IRS variable at clinic $i$ and time step $t$. The random effects are expressed by $(1 + t)U_i$.

$$w(t) = \delta \times \text{rain}_{t-3} + \eta \times \text{rh}_t + \mu \times \text{temp}_{t-1} \tag{8}$$

The matrix of the vector covariates describing weather variables and their corresponding regression coefficients, $w(t)$, includes the fixed effects for rainfall anomalies lagged 3 months ($\delta * rain_{t-3}$), relative humidity anomalies ($\eta * rh_t$), and maximum temperature anomalies lagged 1 month ($\mu * temp_{t-1}$).

## Results

In the uni-variate analysis, a negative binomial model with random intercept for clinic and a random slope for time indicated that IRS suppressed malaria case incidence across the four years (IRR = 0.38, 95% CI: 0.32–0.45). After controlling for seasonal factors and weather anomalies, the effect of IRS is still present. During the months that IRS was implemented with full coverage, monthly malaria incidence is halved across the Illovo estate (IRR = 0.50, 95% CI: 0.42–0.59). Model results are presented in Table 2 with raw coefficients presented in Table D in S1 File.

The conditional modes of random intercept for clinic and the corresponding time slopes for the clinics from the final model are presented in Fig 2. The majority of clinics covered by the IRS programme show higher random intercepts compared to the control clinics. The decrease in time slope for Illovo clinics was generally greater compared to control clinics. Lengwe and Nkombedzi show a different pattern than the other Illovo clinics, with a higher random intercept and a negative time slope.

The plotted time series for each clinic in Fig 3 show the observed malaria cases as points and the model fit as a red line. The correlation between observed and predicted values was 0.90 and the the mean of the residuals was -0.18. The model fit captures the seasonal pattern reasonably well, and mostly captures the effect of the extreme weather in the beginning of 2015. The model fit appears better for the Illovo clinics as compared to the control clinics. The

**Table 2. Selected negative binomial mixed regression model estimates.**

| Co-variate | IRR | 95% CI | | P-value |
|---|---|---|---|---|
| Intercept | 0.03 | 0.02 | 0.05 | <0.001 |
| Time step (month) | 0.98 | 0.97 | 1.00 | 0.024 |
| Sin-term* | 1.00 | 1.00 | 1.00 | <0.001 |
| Cos-term* | 1.00 | 1.00 | 1.00 | <0.001 |
| Rain anomalies lag 3** | 1.09 | 1.02 | 1.17 | 0.018 |
| Relative humidity anomalies*** | 0.75 | 0.66 | 0.84 | <0.001 |
| Maximum temperature anomalies lag 1*** | 0.64 | 0.39 | 1.03 | 0.067 |
| IRS lag 1 | 0.50 | 0.42 | 0.59 | <0.001 |

Results of the final model as presented in Eq 7

* As defined in Eq 6,

**variable re-scaled by factor 100,

***variable rescaled by factor 10.

IRR = incidence rate ratio. CI = Confidence Interval. Time is month of the study period (1–48). IRS is expressed as a proportion of coverage for a specific month and clinic (0-1). Anomalies calculated as difference between value for that month and 20-year average.

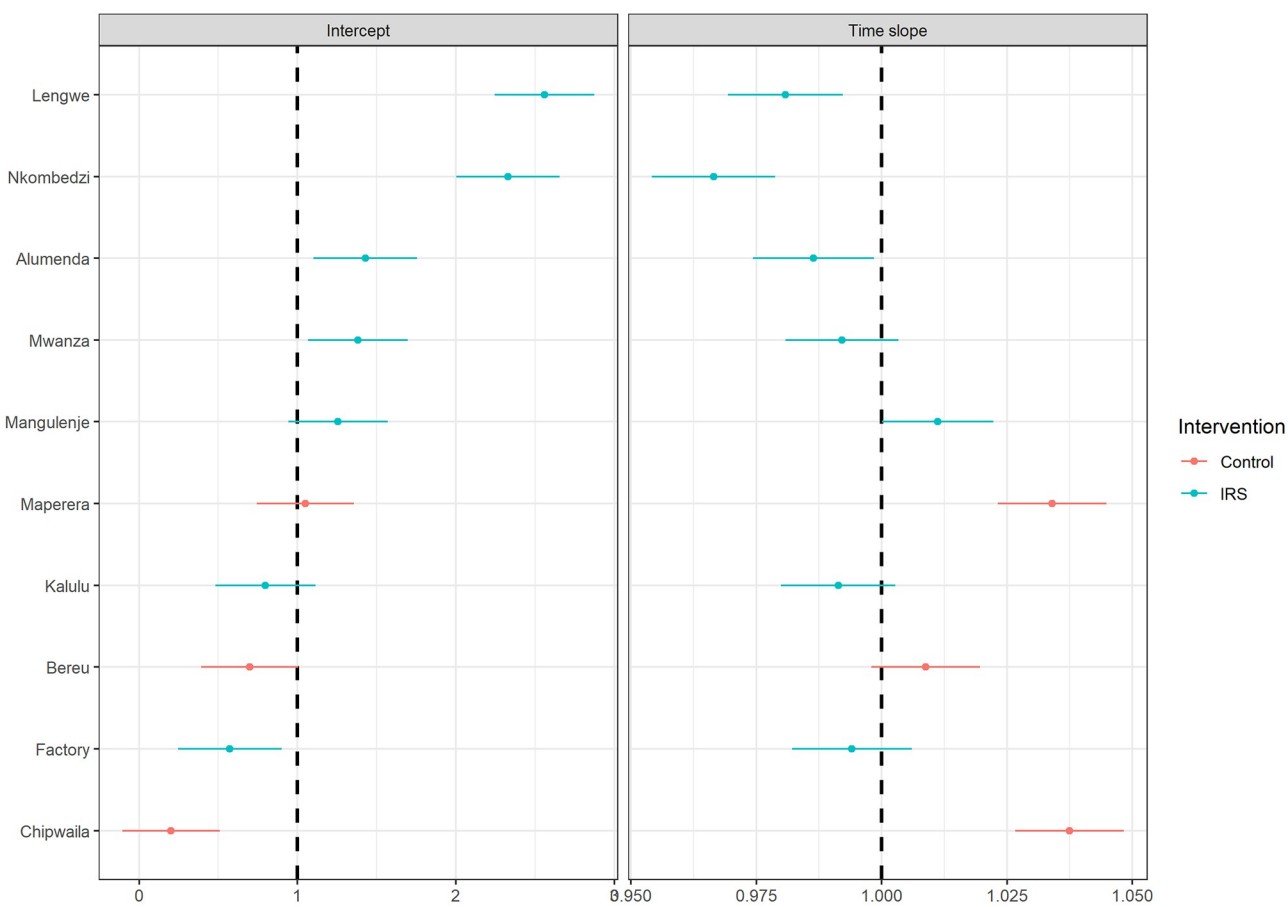

**Fig 2. Conditional modes of the random effects of the fitted model.** The left panel presents the conditional modes of the random intercept for each clinic. The right panel shows the time slope fitted for each of those clinics for a monthly time step. Illovo based clinics that were covered by the IRS programme are in blue and clinics outside of Illovo are in red.

shaded areas that indicate a period of IRS protection, assumed to last 6 months from the start of spraying, often co-occur with periods of lower malaria cases for the Illovo clinics. This period of low incidence is followed by an increase of cases after the protective window of IRS coverage, which could explained by either the impact of IRS, the natural seasonal pattern, or a combination of both. The seasonal pattern for control clinics does not appear vastly different than that of IRS covered clinics. The possibility that the timing of IRS is sub-optimal should be considered. Visualisations of IRS coverage in relation to malaria seasonality are presented in Figs E-G in S1 File to facilitate theoretical discussion on timing and representation of insecticide degradation over time. In Fig 3, similar to Fig 2, a difference in temporal slope can be observed; an increase for control clinics and a decline for most of the Illovo clinics. This is an indication that the long running malaria control programme at Illovo is having an effect.

The average population size over the four year study period residing within the combined catchment areas for IRS-treated clinics was 12,365, whilst the non-intervention clinics served a larger population of 73,622 on average. IRS coverage was defined as the percentage of sprayed households out of the targeted households within catchment areas. Overall coverage was 82%, ranging between 40%-100% for individual clinics in 2015, followed by 86% (54%-92%), 89% (73%-100%), and 73% (12%–81%) in subsequent years, with individual clinics shown in

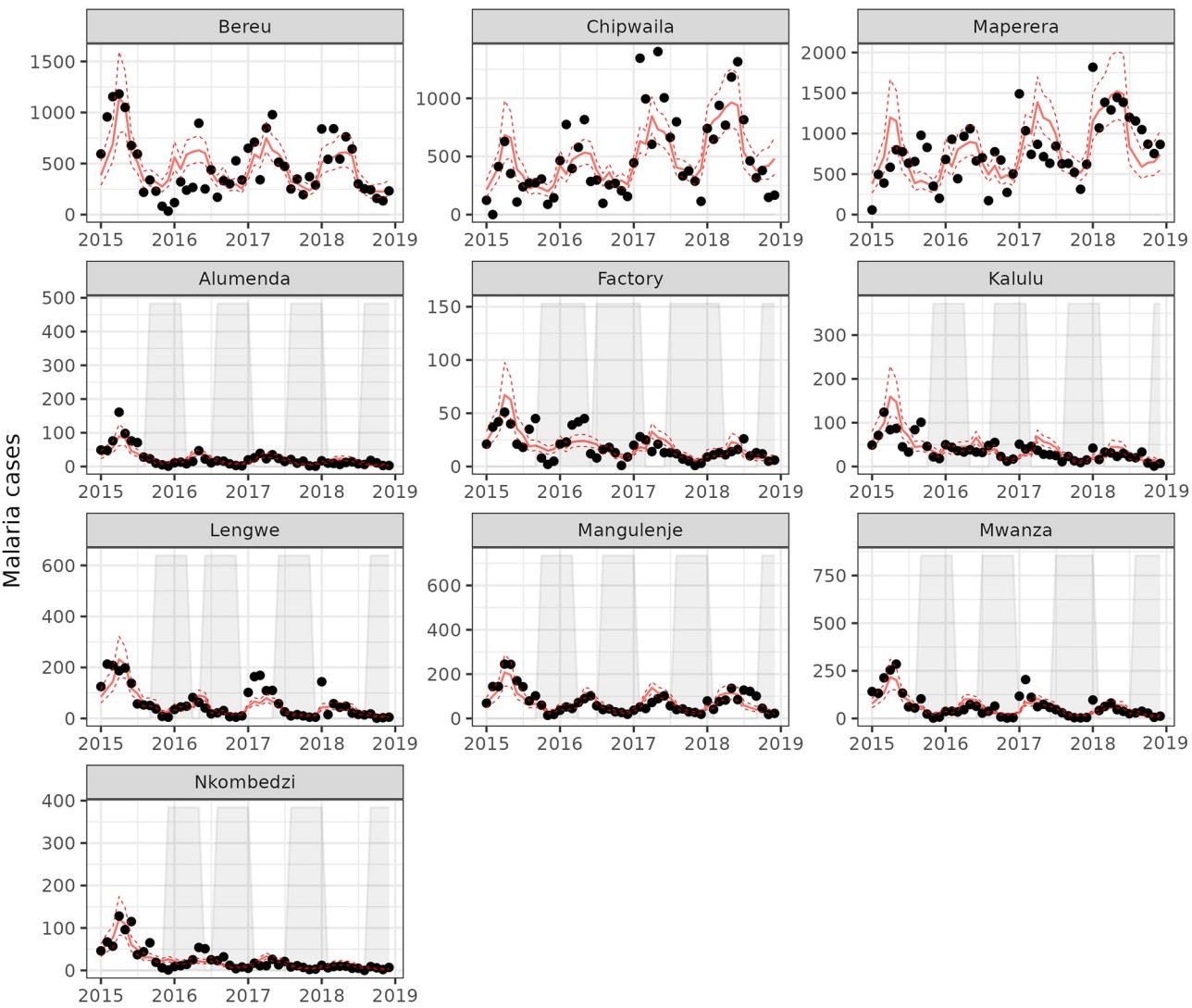

**Fig 3. Time series of malaria cases (black) and fitted line and prediction interval (red) per clinic with IRS period (shaded).** Time series of observed monthly malaria cases are plotted as black points. The fitted line is shown in red. Periods of 6 months duration after the start of IRS are shaded in light-grey. Top row indicates the control clinics, while the rest of the panels received IRS. Prediction intervals were bootstrapped and indicate 2.5th and 97.5th percentiles from a 1000 simulations. Note that the scales on the y-axis are not equal between panels.

Table 3. A large amount of rainfall occurred towards the end 2014 and early 2015, with another intense rainy season occurring in 2017.

A median monthly incidence of 21.51 cases per 1000 at risk (IQR: 10.69–45.80) was reported for the control clinics, versus 14.38 (IQR: 7.62–30.90) for the population targeted by IRS. Further breakdown of case incidence per clinic and year can be found in Table C in S1 File. Fig 4 shows case incidence for both IRS protected and unprotected populations on the left axis, with the exposure as the percentage of IRS covered households displayed on the right axis, represented in shaded gray. This is based on the assumption that IRS with Actellic 300CS is effective for 6 months [14, 15]. The last spray round in 2014 was done between September and November using pyrethroids, which are assumed to be effective for around 3 months or less. Due to the detection of pyrethroid resistance at Illovo the period of protections was likely

**Table 3. IRS coverage and start dates per clinic catchment area for each year 2015–2018.**

| Clinic | 2015 | | 2016 | | 2017 | | 2018 | |
|---|---|---|---|---|---|---|---|---|
| | Coverage | Start date | Coverage | Start date | Coverage | Start date | Coverage | Start date |
| Alumenda | 88% | 11/09 | 88% | 03/08 | 83% | 01/08 | 81% | 01/08 |
| Factory | 40% | 13/10 | 54% | 04/07 | 73% | 04/07 | 12% | 10/08 |
| Kalulu | 63% | 16/10 | 87% | 11/08 | 93% | 09/08 | 79% | 08/10 |
| Lengwe | 100% | 23/09 | 99% | 08/06 | 100% | 23/05 | 96% | 30/08 |
| Mangulenje | 88% | 07/10 | 93% | 21/07 | 83% | 18/07 | 68% | 27/09 |
| Mwanza | 98% | 04/09 | 92% | 16/06 | 94% | 07/06 | 72% | 13/08 |
| Nkombedzi | 95% | 13/11 | 92% | 18/07 | 94% | 11/07 | 72% | 14/09 |
| Overall | 82% | 04/09–13/11 | 86% | 16/06–11/08 | 89% | 23/05–09/08 | 73% | 01/08–08/10 |

Coverage is defined as sprayed households / targeted households within clinic catchment area. Clinic catchment areas are defined as follows: Alumenda (Alumenda, Alumenda Seniors), Factory (Factory, Mess / Riverside, Bonksville, B Compound), Kalulu (Kalulu), Lengwe (Lengwe, Sande Ranch), Mangulenje (Mangulenje, Mlambe, Paxman, Mangulenje Senior), Mwanza (Mwanza, Mechanical Pool / Old School), Nkombedzi (Nkombedzi, Post Office, D Compound). Start date is defined as first date one of the villages within the catchment area has been sprayed.

very short and they are therefore not depicted in Fig 4 [14, 32]. After the heavy rainfall in 2015 incidence is initially higher on the Illovo estate compared with the off-site untreated villages. During the time covered by the 2015 spray round, incidence is similar both in and outside the estate. From 2016 onwards, however, incidence drops in the IRS group compared to the non-IRS group, and this pattern is consistent throughout the 2017 and 2018 spray rounds.

## Discussion

This study provides evidence that repeated annual rounds with Actellic 300CS is effective at reducing routinely reported malaria case incidence rates during an assumed protective IRS

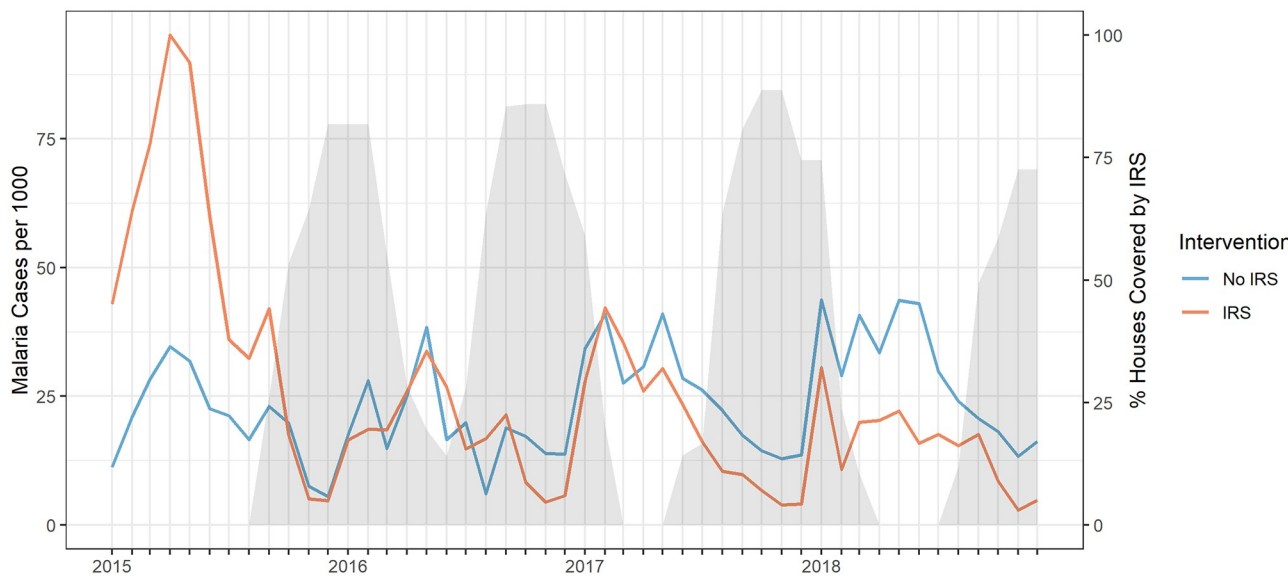

**Fig 4. Malaria case incidence for IRS and non-IRS treated catchment areas.** Monthly malaria cases per 1000 in clinic catchment areas with IRS (blue) and without IRS (red) on the left y-axis. IRS coverage, assuming IRS is effective for six months, at Illovo Nchalo Estate as percentage of targeted houses covered in shaded grey on the right y-axis.

period of 6 months within a low-lying agricultural environment in Malawi. Actellic 300CS is a viable alternative for the Illovo IRS programme, in an agricultural setting with pyrethroid resistance and good ITN coverage [32]. IRS implementation appears to start too early to provide full coverage over the period of high malaria incidence it is intended to provide protection for, as seen in Figs 3 and 4. Additional malaria control measures are recommended to bridge the gap between IRS rounds.

In neighbouring Zambia similar impacts of Actellic 300CS on malaria outcomes have been found by several studies. Hast et al. reported a prevalence rate ratio (PRR) of 0.72 (CI: 0.62, 0.84), over three rounds of Actellic 300CS between 2012–2017. IRS coverage within the study area was approximately 54% with only moderate reductions in *An. gambiae* and *An. funestus* household densities [47, 48]. Furthermore, an effect of IRS on PRR was only observed within the targeted areas during the 6 months after the intervention, that coincided with the rainy season, similar to the time frame in our study [47]. Less pronounced reductions were found by Keating et al. comparing historical malaria incidence between clinics that received IRS with Actellic 300CS in a pyrethroid resistant area, with an IRR of 0.91 (CI: 0.84–0.98) [49]. Although the modeling approach was similar to that used here, the authors assumed the effect of IRS was present throughout the one year study period, which may have resulted in a higher IRR. Similarly in Zambia in 2016, in an area with good ITNs coverage, the incremental protective efficacy (IPE) of pirimiphos-methyl CS compared to the before spray period was not significant beyond 6 months and showed the largest impact between 1–3 months (IPE:0.63, CI: 0.57–0.69) [50]. In western Kenya, a single round of Actellic 300CS resulted in reduced test-positivity rates among febrile patients from 33.3% to 20.6% (12.7%) in IRS treated areas, whilst non-interventions sites reported a 33.2% to 30.4% (2.8%) reduction [51]. During the post-IRS period (15 months) malaria cases dropped between 44–65% in the sub-country hospitals [51].

The nature of our retrospective study meant not all desired information was available. Socio-economic differences and housing quality between Illovo and surrounding villages could not be accounted for. On-site living employees reside in houses built and maintained by Illovo, which are on average of a higher standard than houses from nearby villages. Smaller catchment areas at Illovo may mean access to care is better and managed in a similar way to each other opposed to the more varied catchment population and management of the control clinics, which is reflected in the model evaluation. Analysis without the inclusion of non-intervention clinics showed a very similar effect size to that of the final model, indicating that it is unlikely that these limitations affect the overall conclusion. Other unmeasured variables that could have influenced malaria incidence levels in the region are behavioural factors such as outdoor sleeping and ITN use, but also the seasonal influx of workers during the harvesting season [52]. Seasonal working travelling to Illovo to work undergo a health check at the start of the their contract which includes taking an mRDT, this may cause a spike in positive mRDTs during harvesting season. No household level data was available on bed net distributions and the analysis presented here is based on an assumption that level of ITN use is similar between Illovo and surrounding villages. Because of the listed limitations it is unlikely that the long-term reduction in malaria IRR is solely attributable to IRS. Despite this uncertainty, the fact that this study compares effect of IRS between time points within IRS covered clinics, as well between IRS and control clinics, does support the idea that adding IRS to ITNs provides additional benefit.

In the case of Illovo Sugar Malawi, a benefit of privately funded IRS is the continuity in funding and training of the spray team since 1990 [24]. By introducing mono-culture and irrigation to an area, large-scale agricultural businesses alter the dynamics of malaria transmission in an area. Irrigation provides a steady year-round water supply, which, depending on whether it's active or passive irrigation, and how well-maintained irrigation channels are, could

increase mosquito breeding site availability. Employees and their families living closely to these breeding sites, could be exposed to more mosquito bites. On the other hand, economic prosperity may negate some of the malaria risk through improved housing and access to care. There is a strong incentive for businesses, especially those hiring for labour-intensive jobs, to promote good health among their employees. This is both a social responsibility and good business sense, as it will lead to less absenteeism, greater worker satisfaction, and therefore increased productivity. There have been multiple examples of companies running malaria control programmes, mostly in the agricultural and mining industry, which in some cases benefit the wider population [53]. To slow the emergence of resistant genotypes within mosquitoes and sugar pests, Illovo has been using a mosaic of insecticides with different modes of action since 2019. A draw-back from privately organised IRS campaigns is the lack of standardised, publicly available data on monitoring and evaluation. Sharing information with the NMCP and scientific community could help in the detection of insecticide resistance, residual effect of insecticides, improve training and reporting practices, and increase accountability of both the NMCP and private companies.

Whilst a systematic review of hut trials shows that the probability of pirimiphos-methyl killing mosquitoes entering the house starts to decline after 6 months, there are not many studies looking at the time-span for which Actellic 300CS provides protection against malaria and which factors affect its effectiveness in the field [14, 21]. Our study reports effective protection for an assumed period of 6 months, but finer-scale temporal data is needed to provide more insight. Field reports from PMI Malawi suggest a residual life of 2–5 months, much lower than effectiveness measured in experimental studies, which could be explained by environmental and housing factors influencing degradation of the insecticide, or IRS application procedures [25]. Additionally, the protracted implementation period of the IRS campaign could have resulted in different windows of protection between villages, potentially offering sub-optimal protection during the malaria season in some villages and the necessary protection in others [21]. How exactly this has affected the estimation of IRS effectiveness is difficult to quantify due to the necessary data aggregation from village to clinic level. While the results show that periods of lower malaria incidence coincide with periods of IRS, the natural seasonality pattern could also have contributed to the observed IRS effect estimate, although the inclusion of control clinics mitigate this concern somewhat. Improved reporting by IRS campaigns on spray quality indicators, insecticide resistance, and both entomological and disease outcomes at regular time intervals, could provide the data necessary to uncover the reasons why IRS with Actellic 300CS performance is inconsistent across settings. The timing of IRS and the duration of the protective period should not be overlooked in the planning of IRS and trials assessing interventions. Studies have shown, consistent with results presented here, that there can be a strong rebound effect at the end of the protective period [54].

## Conclusion

Within an agricultural, low-lying area of malaria, where pyrethroid resistance has been reported in malaria vectors, IRS with Actellic 300CS significantly contributes to lowering malaria incidence. The implementation of annual spraying with pirimiphos-methyl (Actellic 300CS) over the years 2015–2018 has reduced monthly malaria incidence by approximately half during the protective period compared to time periods outside of the protective period and compared to control clinics. This study is an example of how privately funded IRS programmes can contribute to malaria control and the health of employees. Further research is needed on the optimal temporal coverage and timing of IRS in combination with other malaria control tools to maximise benefits.

## Supporting information

**S1 File. Supplementary materials.** The S1 File contains additional information on methodological choices made throughout the research process, as well as additional data summaries to improve understanding of the dataset.

List of included tables:

S1 Table A. Characteristics of control clinics used for their selection.

S1 Table B. Median malaria cases per year and per clinic.

S1 Table C. Unadjusted incidence rates between IRS and non-IRS treated clinics per year.

List of included figures:

S1 Fig A. Weather patterns over time.

S1 Fig B. Fitted harmonic regression line vs observed data points.

S1 Fig C. Auto-Correlation Function (ACF) plot for 5 clinics.

S1 Fig D. Partial Auto-Correlation Function (PACF) plot for 5 clinics.

S1 Fig E. Illustration of IRS coding per clinic—not taking into account insecticidal degradation.

S1 Fig F. Illustration of IRS coding per clinic—assuming linear degradation of the insecticide.

S1 Fig G. Illustration of IRS coding per clinic—assuming exponential degradation of the insecticide.
(PDF)

## Acknowledgments

The author would like to acknowledge Dr. Julie-Anne Tangena for her valuable feedback.

## Author Contributions

**Conceptualization:** Remy Hoek Spaans, Christopher M. Jones, Michelle C. Stanton.

**Data curation:** Remy Hoek Spaans, Peter Nasoni.

**Formal analysis:** Remy Hoek Spaans.

**Funding acquisition:** Remy Hoek Spaans.

**Investigation:** Remy Hoek Spaans.

**Methodology:** Remy Hoek Spaans.

**Project administration:** Remy Hoek Spaans.

**Resources:** Albert Mkumbwa, Peter Nasoni, Christopher M. Jones.

**Supervision:** Christopher M. Jones, Michelle C. Stanton.

**Writing – original draft:** Remy Hoek Spaans.

**Writing – review & editing:** Christopher M. Jones, Michelle C. Stanton.

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
