## [Decision Letter · Decision Letter 0]

2 Oct 2023

PGPH-D-23-01375

Impact of four years of annually repeated indoor residual spraying (IRS) with Actellic 300CS on routinely reported malaria cases in an agricultural setting in Malawi

Dear Dr. Hoek Spaans,

Thank you for submitting your manuscript to PLOS Global Public Health. After careful consideration, we feel that it has merit but does not fully meet PLOS Global Public Health’s publication criteria as it currently stands. Therefore, we invite you to submit a revised version of the manuscript that addresses the points raised during the review process.

The manuscript has been evaluated by two reviewers, and their comments are available below. Reviewer #1 responded positively to your manuscript. Reviewer #2 raised several concerns which, if addressed, will improve your submission. They have requested that you provide the original dataset and analysis code. They suggest amending your discussion to consider factors that may affect your data analysis such as the waning effect of intervention and the time of year. Based on these points, they also suggest re-analysing the data. They recommend edits to your Methods and Results which may improve clarity and readability. Could you please revise the manuscript to carefully address the concerns raised.

We look forward to receiving your revised manuscript.

Kind regards,

Richard Ali

Staff Editor

Journal Requirements:

1. Please note that PLOS ONE has specific guidelines on code sharing for submissions in which author-generated code underpins the findings in the manuscript. In these cases, all author-generated code must be made available without restrictions upon publication of the work. Please review our guidelines at https://journals.plos.org/plosone/s/materials-and-software-sharing#loc-sharing-code and ensure that your code is shared in a way that follows best practice and facilitates reproducibility and reuse.

a. State what role the funders took in the study. If the funders had no role in your study, please state: “The funders had no role in study design, data collection and analysis, decision to publish, or preparation of the manuscript.”

b. If any authors received a salary from any of your funders, please state which authors and which funders.

3. Please send a completed 'Competing Interests' statement, including any COIs declared by your co-authors. If you have no competing interests to declare, please state "The authors have declared that no competing interests exist". Otherwise please declare all competing interests beginning with twhe statement "I have read the journal's policy and the authors of this manuscript have the following competing interests:"

4. We ask that a manuscript source file is provided at Revision. Please upload your manuscript file as a .doc, .docx, .rtf or .tex.

5. Some material included in your submission may be copyrighted. According to PLOS’s copyright policy, authors who use figures or other material (e.g., graphics, clipart, maps) from another author or copyright holder must demonstrate or obtain permission to publish this material under the Creative Commons Attribution 4.0 International (CC BY 4.0) License used by PLOS journals. Please closely review the details of PLOS’s copyright requirements here: PLOS Licenses and Copyright. If you need to request permissions from a copyright holder, you may use PLOS's Copyright Content Permission form.

Potential Copyright Issues:

Fig 1: please (a) provide a direct link to the base layer of the map (i.e., the country or region border shape) and ensure this is also included in the figure legend; and (b) provide a link to the terms of use / license information for the base layer image or shapefile. We cannot publish proprietary or copyrighted maps (e.g. Google Maps, Mapquest) and the terms of use for your map base layer must be compatible with our CC-BY 4.0 license. 

"

Additional Editor Comments (if provided):

Reviewers' comments:

Reviewer's Responses to Questions

**Comments to the Author**

1. Does this manuscript meet PLOS Global Public Health’s publication criteria? Is the manuscript technically sound, and do the data support the conclusions? The manuscript must describe methodologically and ethically rigorous research with conclusions that are appropriately drawn based on the data presented.

Reviewer #1: Yes

Reviewer #2: Partly

2. Has the statistical analysis been performed appropriately and rigorously?

Reviewer #1: Yes

Reviewer #2: Yes

3. Have the authors made all data underlying the findings in their manuscript fully available (please refer to the Data Availability Statement at the start of the manuscript PDF file)?

Reviewer #1: Yes

Reviewer #2: No

4. Is the manuscript presented in an intelligible fashion and written in standard English?

Reviewer #1: Yes

Reviewer #2: Yes

5. Review Comments to the Author

Reviewer #1: A very well written paper that examines some assumptions and shows the value of private sector investment in IRS. Despite the limitations of retrospective data, the authors do a thorough analysis and provide useful insights and conclusions. I don't have any editorial comments or suggestions. It was a well done piece of work.

Reviewer #2: Please refer to the attached feedback (repeated here for convenience)

This is an interesting study that considers the impact of spraying Actellic 300CS indoors in people’s homes on malaria incidence within a zone of privately managed sugar plantation. The study compares outcomes from 7 clinics serving the residential community within the plantation to data from 3 health facilities serving people living outside the plantation. The main finding reported is that the repeated IRS application does have benefit to the community of permanent residents in the plantation zone. It is a useful study particularly because little information is often available on privately managed spray campaigns, so this study adds insight to these types of operations.

The paper is well-contextualised given previous literature that considers research and policy recommendation efforts by the wider community.

Scientific comments / concerns

Are original data deposited in appropriate repositories? The original data and analysis code are not part of the original submission (summary data are noted in the supplement), please share these or share the repository for these if possible.

Data concerns:

The IRS cover is estimated for each village and then all villages that are serviced by a given clinic are aggregated to estimate a clinic level IRS cover metric. The cover is determined to be the proportion of sprayed houses of all targeted houses and set at this value from the final month of the spray team delivering the intervention to that village, for the next 6 months. What about the waning efficacy of the intervention? This can be quite different in different locations for various reasons that the authors note. Would this approach not over-estimate cover for patients particularly toward the later months of a campaign? I think the analysis would likely still find a positive benefit from IRS, but it would certainly be worth commenting on this in the discussion. This is especially important because the authors note that IRS could be sprayed closer to the transmission season in the discussion. Please comment more on the assumptions of this framework. Potentially, you could try incorporating some waning e.g. Opiyo et al 2022?

To expand on this a little bit, the incremental spraying period is relatively prolonged (May to November) – how did you account for the differential protective window across the population (see Opiyo et al 2022 for thoughts on this stepped protection)? Some households would likely be unprotected by November, when others are first gaining protection. The level of community protection would therefore be quite heterogeneous over the year. How did this incremental spraying change for 2015-2018? Could this have any effect on the transmission patterns observed?

The 2015 year was noted to be particularly unusually wet, and lost benefit from the IRS as pyrethroid-products were being used. Would this not skew the results to over-estimate the Actellic effect? The 2015 was scored to have IRS cover as 0 for all clinics, and as this is when the incidence in the plantation is higher than elsewhere, the following effects from the IRS cover > 0 variable would lead to higher IRR (I think?) Is this fair? Could you repeat the analysis starting later in that first year to avoid this potential problem? Or comment on this in the interpretations?

The methods indicate that “the previous month’s malaria cases were added to the model as an auto-regressive term to adjust for potential temporal variation.” Could this lead to over fitting if you also are including the temporal trend?

At the editors discretion, it might improve readability if the model selection section was shifted to methods and only the results from Model 5 were then presented in the results. There is some repetition in the results section (e.g. the start of paragraph after Table 2) that can be cut.

It would be useful to present the raw data and code or at least the coefficients and intercepts of the modelling process, at least in the supplement, allowing the reader to reproduce results shown in Table 3 (and look at the change in probability of cases rather than IRR which is sometimes a preferred method to translate the findings from these types of models e.g. see Andrew Gelman et al. Regression and other stories - book).

The major conclusion is that Actellic IRS provides protection. Please could you comment on integrated resistance management. What sort of monitoring is needed to ensure resistance does not develop to the organophosphate product – could you add some thoughts on when this product should be rotated?

The timing of the most recent ITN campaigns is likely important. The assumption is that it is equivalent for people within and outside the plantation.

Minor

Introduction

The statement in the introduction that “Anopheles spp are shifting towards more exophagic behaviour” is possibly a little strong. The referenced papers are indicative … perhaps soften and say “with evidence suggesting that Anopheles spp may be shifting …”

There is a reference for “adaptations to the house after spraying” (Opiyo et al 2022 DOI: 10.1371/journal.pgph.0000227)

Results

The mean monthly incidence for cases per 1000 at risk are slightly different in the main manuscript and supplement (only at the second decimal place) but perhaps best to align these.

In the results section, there is a sentence at lines 206-208 that does not make sense “For both treatment groups, the annual incidence pattern mimics the weather patterns from with a lag of 1-3 months”. Please rephrase. At this section, the results state the lag for the incidence from the wea

---

## [Decision Letter · Decision Letter 1]

12 Jan 2024

PGPH-D-23-01375R1

Impact of four years of annually repeated indoor residual spraying (IRS) with Actellic 300CS on routinely reported malaria cases in an agricultural setting in Malawi

Dear Dr. Hoek Spaans,

Thank you for submitting your manuscript to PLOS Global Public Health. After careful consideration, we feel that it has merit but does not fully meet PLOS Global Public Health’s publication criteria as it currently stands. Therefore, we invite you to submit a revised version of the manuscript that addresses the points raised during the review process.

While all previous feedback has been adequately addressed, Reviewer #2 has identified a few remaining points that require clarification.

We look forward to receiving your revised manuscript.

Kind regards,

Ruth Ashton, Ph.D.

Academic Editor

Journal Requirements:

1. Please note that PLOS ONE has specific guidelines on code sharing for submissions in which author-generated code underpins the findings in the manuscript. In these cases, all author-generated code must be made available without restrictions upon publication of the work. Please review our guidelines at https://journals.plos.org/plosone/s/materials-and-software-sharing#loc-sharing-code and ensure that your code is shared in a way that follows best practice and facilitates reproducibility and reuse."

2. We would like to request for copy editing. 

Additional Editor Comments (if provided):

Reviewers' comments:

Reviewer's Responses to Questions

**Comments to the Author**

1. If the authors have adequately addressed your comments raised in a previous round of review and you feel that this manuscript is now acceptable for publication, you may indicate that here to bypass the “Comments to the Author” section, enter your conflict of interest statement in the “Confidential to Editor” section, and submit your "Accept" recommendation.

Reviewer #1: All comments have been addressed

Reviewer #2: All comments have been addressed

2. Does this manuscript meet PLOS Global Public Health’s publication criteria? Is the manuscript technically sound, and do the data support the conclusions? The manuscript must describe methodologically and ethically rigorous research with conclusions that are appropriately drawn based on the data presented.

Reviewer #1: Yes

Reviewer #2: Yes

3. Has the statistical analysis been performed appropriately and rigorously?

Reviewer #1: Yes

Reviewer #2: Yes

4. Have the authors made all data underlying the findings in their manuscript fully available (please refer to the Data Availability Statement at the start of the manuscript PDF file)?

Reviewer #1: Yes

Reviewer #2: Yes

5. Is the manuscript presented in an intelligible fashion and written in standard English?

Reviewer #1: Yes

Reviewer #2: Yes

6. Review Comments to the Author

Reviewer #1: The paper is well written and the various questions from reviewers have been addressed adequately.

Reviewer #2: Thank you for the revised version to review. There are only minor considerations left in my opinion that would be useful to clarify.

I am somewhat confused by the discussion of the results of the modelling. The random intercepts and slopes for the clinics are noted to be unusually high for Lengwe and Nkombedzi. The authors state this is due to (lines 225-226) “unmeasured covariates varying at clinic level”. But are these intercepts simply capturing the relative burden of cases at t = 0? So, the interpretation is that Lengwe and Nkombedzi have higher cases relative to other clinics at this time? Both of these locations have relatively high and negative slopes indicating that, over the period studied, there has been reductions in cases. I am unsure how this demonstrates “unmeasured covariates”? I think simply rephrasing this would be a good idea. There will always be uncertainty that we are not capturing with models particularly when fitted to ecological data.

I think there is an issue with Figures referenced in the text and attached. I think the Figure 4 legend reads as though it is describing Figure 2? Please carefully check throughout.

The results refer to Fig 2 when I think you mean Fig 3? At line 223. But please see the note above as I think it is a mismatch with uploaded figures.

Can you also show some uncertainty on the plots for Figure 3 (sampling from the posterior predictions) – these are very interesting and worth noting the uncertainty.

Page 6 line 37, says “… 2 to 9 months or residual activity…”, please correct to of

It is noted that 3 clinics were selected for control states although others fitted the criteria. Out of interest, was it not possible to include all these to gain inference from more data? Even if the areas were capturing larger populations? There is no need to do anything in this publication but if the data are available, why not use them?

7. PLOS authors have the option to publish the peer review history of their article (what does this mean?). If published, this will include your full peer review and any attached files.

**Do you want your identity to be public for this peer review?** For information about this choice, including consent withdrawal, please see our Privacy Policy.

Reviewer #1: No

Reviewer #2: **Yes: **Ellie Sherrard-Smith

---

## [Editor Report · Decision Letter 2]

3 Apr 2024

Impact of four years of annually repeated indoor residual spraying (IRS) with Actellic 300CS on routinely reported malaria cases in an agricultural setting in Malawi

PGPH-D-23-01375R2

Dear Ms Hoek Spaans,

We are pleased to inform you that your manuscript 'Impact of four years of annually repeated indoor residual spraying (IRS) with Actellic 300CS on routinely reported malaria cases in an agricultural setting in Malawi' has been provisionally accepted for publication in PLOS Global Public Health.

Best regards,

Ruth Ashton, Ph.D.

Academic Editor